# Role of the Monocyte–Macrophage System in Normal Pregnancy and Preeclampsia

**DOI:** 10.3390/ijms20153695

**Published:** 2019-07-28

**Authors:** Polina Vishnyakova, Andrey Elchaninov, Timur Fatkhudinov, Gennady Sukhikh

**Affiliations:** 1National Medical Research Center for Obstetrics, Gynecology and Perinatology Named after Academician V.I. Kulakov of Ministry of Healthcare of Russian Federation, 4 Oparina Street, 117997 Moscow, Russia; 2Peoples’ Friendship University of Russia, 6 Miklukho-Maklaya Street, 117198 Moscow, Russia; 3Scientific Research Institute of Human Morphology, 3 Tsurupa Street, 117418 Moscow, Russia

**Keywords:** preeclampsia, monocyte, macrophage, placenta, decidua, inflammation

## Abstract

The proper functioning of the monocyte–macrophage system, an important unit of innate immunity, ensures the normal course of pregnancy. In this review, we present the current data on the origin of the monocyte–macrophage system and its functioning in the female reproductive system during the ovarian cycle, and over the course of both normal and complicated pregnancy. Preeclampsia is a crucial gestation disorder characterized by pronounced inflammation in the maternal body that affects the work of the monocyte–macrophage system. The effects of inflammation at preeclampsia manifest in changes in monocyte counts and their subset composition, and changes in placental macrophage counts and their polarization. Here we summarize the recent data on this issue for both the maternal organism and the fetus. The influence of estrogen on macrophages and their altered levels in preeclampsia are also discussed.

## 1. Introduction

The prevalence, main symptoms, and classification of preeclampsia (PE) are well established and can be found in every article devoted to this multisystem pregnancy complication. Hundreds of studies devoted to the cell and animal models of PE, as well as biological samples from patients with PE are published every year. However, the scientific community is still wondering what are the main causes of PE and is it possible to predict and prevent the development of this widespread pregnancy disorder?

It is now clear that PE is a multifactorial syndrome, but not an isolated disease. PE occurs in the second half of pregnancy (after the 20th week) and is characterized by arterial hypertension (depending on the severity of PE) in combination with proteinuria (≥0.3 g/L in daily urine) and/or manifestations of multi-organ or multisystem dysfunction/failure [1]. The frequency of PE depends on the country; it is estimated at 8% on average [2]. PE is associated with insufficiently deep placentation, which may be associated with the impairment of spiral arteries remodeling and the presence of obstructive injuries in myometrium. PE is characterized by systemic immune activation associated with elevated levels of inflammatory cytokines produced by various cell types in blood and tissues [3].

Today, a large amount of accumulated data suggests that the dysfunctional maternal immune response in the mother’s organism at PE is manifested by altered functional activity of monocyte–macrophage system, which is the most important unit of innate immunity. According to the modern classification, there are three groups of monocytes—classical, intermediate, and non-classical. Each subpopulation has its own function and characteristic markers, as detailed below. Tissue macrophages are usually divided into proinflammatory (M1) and anti-inflammatory (M2), although the distinction between these types is currently being revised. In this regard, several questions remain open, notably which subpopulations of monocytes are predominantly destined to become M1- or M2-polarized macrophages and whether identical monocytes from the same subpopulation can undergo differential polarization in tissues. It is now known that polarization is triggered by local concentrations of certain cytokines. However, it is not clear whether selective depletion of a particular monocyte population will affect the composition of tissue macrophage populations. The answers will provide a relevant support to identification of early predictors for PE.

This article attempts to link the knowledge on developmental origins of the monocyte–macrophage system to the ways of its functioning during normal pregnancy and in PE. Having addressed the key differential characteristics of monocytes we ultimately discuss their possible relevance as predictors in the prevention and diagnosis of PE. The review is focused on studies published over the last few years in order to provide the most up-to-date information on the topic. 

## 2. Monocyte–Macrophage System during Pregnancy

### 2.1. Monocyte–Macrophage System: Developmental Origins and Cell Lineage

Macrophages play a key role in the maintenance of tissue homeostasis, regulation of inflammatory processes and tissue repair. In accordance with modern concepts, tissue macrophages in mammalian ontogenesis develop from three sources that correspond to three generations of hematopoietic progenitor cells [4,5].

The first generation of hematopoietic cells is specified within the wall of the yolk sac. It is important to note that these hematopoietic cells have a different origin than progenitor cells developing from hematopoietic islets in the endothelium of yolk sac capillaries [5]. It is supposed that microglial cells of the central nervous system descend from these very first hematopoietic progenitors [6]. Life cycle of a macrophage usually involves a migratory stage represented by monocytes circulating in the blood; this stage is absent in microglia development. Microglial precursors migrate directly to the central nervous system and mature within [6].

The second generation, erythro-myeloid progenitor cells, is formed from the hematogenic endothelium of the yolk sac capillaries. These cells subsequently colonize the embryo’s liver. By the profile of molecular markers, macrophages derived from these progenitors are very similar to macrophages derived from the first generation of progenitors; however, their maturation involves the stage of monocytes [5,6].

The third generation of hematopoietic progenitors develops from the hematogenic endothelium of the aorta–gonad–mesonephros zone; these cells subsequently migrate to the liver and red bone marrow. Macrophages derived from this generation colonize almost all organs of the embryo except the central nervous system [5].

Thus, macrophage populations of most organs in the prenatal period consist of hematopoietic cells descending from the second and third generations. In most of them, the proportion of macrophages descending from erythro-myeloid progenitor cells of the yolk sac is being gradually decreased, and accordingly the proportion of macrophages descending from hematopoietic cells of the third generation is increased [4,5]. However, there are three exceptions. First of them is the central nervous system (which is apparently inaccessible to monocytes and macrophages except for the macrophages of the first generation). The other two are the liver and the epidermis, where only macrophages descending from the second generation of hematopoietic cells are normally found, named Kupffer cells and Langerhans cells, respectively [5]. Eventually, macrophages of embryonic origin (descendants of the second generation of hematopoietic progenitors) completely disappear from connective tissues of skin dermis and intestinal tract mucosae, being replaced by macrophages of bone marrow origin [4,5,7]. The reasons for such particular macrophage distribution within the mammal organism are unknown.

Macrophages of the second and third generation undergo the stage of monocytes in their development. In the postnatal period, based on CD14 (lipopolysaccharide coreceptor (LPS) and CD16 expression patterns (Fc receptor-FcγRIII), there are three subpopulations of blood monocytes: classical (CD14++, about 90%), non-classical (CD16++, about 10%), and a small intermediate population of monocytes expressing high levels of CD14 and CD16 [8]. These subpopulations have different properties. CD14++ monocytes are considered mature; they show pronounced phagocytic activity and are capable of producing reactive oxygen species and cytokines through activation of TLR signaling pathway [9]. CD16++ cells do not produce reactive oxygen species but are better at production of pro-inflammatory cytokines. CD16++ cells are the patrolling monocytes which perpetually assess the state of the endothelium and infiltrate tissues under normal conditions and during inflammatory processes [8,9]. The role of the intermediate population of monocytes is poorly understood, but given the high expression level of MHC-II they probably participate in antigen presentation and activation of T lymphocytes [10].

The data on the ontogenesis of mammalian macrophages were obtained mainly on various lines of laboratory mice. The number of works concerning studies dealing with the development of macrophage populations in humans are few. It is reasonably supposed that the program of tissue macrophage development in humans is generally consistent with that in mice [11,12].

Macrophages are a heterogeneous population of cells, not only in terms of the source of their development but also in their functional characteristics. Macrophages are capable of rapid adaptation by changing their phenotype and functions under the influence of various signaling molecules. Activation of macrophages in situ can be directed towards either pro-inflammatory M1 or anti-inflammatory M2 polarized macrophages, which differ not only by expression of specific markers but also by their roles in immune response [13]. A local shift of the M1/M2 balance towards M2 in the area of damage significantly improves the dynamics and efficiency of reparative processes; it has been convincingly demonstrated for skin wounds [14], spinal cord injuries [15], myocardial infarction, and cardiomyopathy [16] among other models. Macrophages could be polarized using specific inductors (what is often called the direct polarization), or using an indirect method: blockage the undesired phenotype and get the reverse, desired, phenotype. At the same time, a part of macrophages may remain in a non-activated state [17]. Thus, the in vivo M2 phenotype could be achieved by blocking IL-6-signaling [15] or by adding IL-4 [18] and IL-10 what induced М2а and M2c polarization, respectively [13]. However, it is often emphasized that phenotypes of activated macrophages should be seen as a continuum, with M1 and M2 being its extremum variants [19]. The common markers of M1 macrophages are CD80/86, CD11c, and iNOS, whereas M2 phenotype is usually characterized by expression of CD163, CD206, and Arginase 1 (Arg1) [20,21]. At the same time, CD68- and CD14-specific staining is a common approach for identification of total macrophage population in tissue and the number of positive cells often serves as a normalization value for M1 and M2 cell counts.

### 2.2. The Role of Monocytes in Pregnancy

During normal pregnancy, an increase in the number of blood monocytes and their activation are observed. These events are accompanied by a change in the ratio of blood monocyte subpopulations: an increase in the number of intermediate monocytes with high levels of CD14 and CD16, and a decrease in the number of the classical monocyte population (CD14++) [22]. Another study revealed an increase in the number of classical monocytes and a decrease in the number of non-classical monocytes [23].

The high heterogeneity of monocyte populations is well known; however, most of the studies have been performed on CD14++ monocytes. An increase in CD11b, CD14, and CD64 monocyte markers in the blood of pregnant women is observed along with high levels of the oxygen free radical production and a decrease in phagocytic activity [24]. The data on cytokine production by unstimulated monocytes are controversial, which may reflect the influence of methods used for monocyte isolation. However, LPS stimulation promotes a decrease in cytokine production in the blood of pregnant women as compared to non-pregnant [25,26]. Another shortcoming of the studies on monocytes in pregnancy is related to the term of gestation: most of the studies have been carried out on blood monocytes collected at the third trimester of pregnancy, the corresponding data for other terms are largely missing [25,26].

The exact mechanisms of monocyte activation during pregnancy are unknown. It is assumed that the placenta plays a leading role in this process. Monocytes, circulating with the blood through placental lacunae, come into contact with syncytiotrophoblast which can activate them towards pro-inflammatory phenotype [27,28]. In addition to the direct contacts, monocytes can be activated indirectly by cytokines [29,30], by microvesicles and exosomes released from syncytiotrophoblast into the maternal blood [31,32,33,34] and by pregnancy hormones, e.g., estrogens. A number of studies indicate that estrogens exert anti-inflammatory effect on monocytes [35,36,37,38]. Estrogens downregulate the expression of chemokine receptors CCR2 and CXCR3 and suppress the monocyte migration capacity evoked by MIP-1α and MCP-1/JE stimulation [35,36,39]. Estrogens also downregulate production of IL-1 by LPS-stimulated monocytes [40]. In blood, high levels of 17-estradiol are associated with increased numbers of monocytes expressing the markers of M2 macrophages [41]. The anti-inflammatory effect of estrogens on monocytes is believed to be mediated by a specific splice variant of the ERα36 receptor [42]. In vitro, estrogens induce monocyte apoptosis by increasing FasL expression [43]. There is still no data on the possible influence of preeclamptic hypoestrogenemia on the counts and population structure of monocytes. However, it has been established that reduced concentrations of 17-estradiol facilitate the expression of CD16 and boost the production of pro-inflammatory cytokines TNFalpha (Tumor necrosis factor), IL-1, and IL-6 [37].

### 2.3. The Role of Macrophages in Female Reproductive System Prior to and during Pregnancy

Macrophages are found in all organs of the female reproductive system, their populations being represented by both the monocyte-derived macrophages and the resident macrophages that colonize organs in the prenatal period [44]. Macrophages are unevenly distributed in the endometrium; their numbers and density vary depending on the stage of the menstrual cycle. Within the endometrium, several macrophage populations are distinguished. One of them is located closer to the uterine lumen and is supposed to be involved in the processes of desquamation and regeneration; another population is found mainly around the uterine glands [45]. During the proliferative phase, endometrial macrophages express surface proteins (Transferrin receptor protein 1 (TFRC), CD69 and intracellular adhesion molecule 1), matrix remodeling factors, cytokines, and growth factors that prepare endometrium for possible implantation or the induction of desquamation [45].

During the proliferative phase, macrophages are located within the stroma of superficial layer of endometrium, surrounding and penetrating the lumens of uterine glands. During the secretory phase, the number of macrophages in the endometrium increases dramatically [46]. It is shown that the number of CD14+ macrophages increases by about 45% [47]. During the proliferative phase and in the beginning of the secretory phase, the number of macrophages increases due to proliferation of the resident macrophages; at the end of the secretory phase, migration of monocyte–macrophages to endometrium is observed. It is assumed that during desquamation endometrial macrophages partially migrate from endometrium to lymph nodes or die by apoptosis [48]. In the case of fertilization, the trophoblast invasion occurs at the site of placentation accompanied by accumulation of macrophages in decidua, the pregnancy-modified endometrium. The main functions of decidual macrophages are secretion of cytokines and growth factors for successful placentation, providing immune tolerance to the semi-allogeneic fetus and protection of the fetus against infections. Decidual macrophages mostly originate from monocytes circulating in the blood. Macrophages that reside in the placenta amount to no less than 20%–30% of the total macrophages in the body; they play a key role in the establishment of the immunological aspects of mother–fetus interaction [49,50]. Remodeling of the spiral arteries in maternal uterus is also supported by local decidual macrophages. The involvement of macrophages in the remodeling of spiral arteries is determined by the fact that macrophages secrete many factors universally involved in angiogenesis and tissue remodeling [51,52]. Angiogenic factors secreted by decidual macrophages include angiogenin, keratinocyte growth factor, fibroblast growth factor B, vascular endothelial growth factor A, and angiopoietins 1 and 2. Remodeling factors synthesized by decidual macrophages include matrix metalloproteinases 1, 2, 7, 9, and 10 [52,53]. At the same time, the high phagocytic activity of decidual macrophages is indispensable for the uptake of dead cells that undergo apoptosis during the remodeling of spiral arteries and decidual membrane. The timely disposal of apoptotic cells has been shown to prevent the risks of endothelium activation and excessive attraction of monocytes [54]. Modulation of immune reactions occurs in the placenta throughout pregnancy with macrophages playing a central role in this process [55]. Placentation in the first and second trimesters of pregnancy is characterized by pro-inflammatory environment (favoring M1 polarization), which ensures the correct restoration of the uterine epithelium and protection against infections. The second and third trimesters are the periods of rapid growth of the fetus at the advanced stages of its development, and the prevailing immunological profile is anti-inflammatory (favoring the alternative M2 polarization). A similar immunological shift is observed for other immune cell types including T helper 2 cells (Th2) and a subset of suppressor CD4+ T cells (regulatory T cells, Treg). Th2 and Treg cells are responsible for maintaining peripheral immune tolerance during pregnancy [56]. Successful course of pregnancy is accompanied by activation of anti-inflammatory Th2 and a decrease in Th1/Th2 cytokine ratio [57,58,59]. Treg cells play a principal role in the protection against recognition of semiallogeneic fetus by the immune system of maternal organism. Deviations in Treg cell counts are associated with different pregnancy complications [60]. During delivery, the immune profile returns to pro-inflammatory state which is necessary for the uterine contraction and fetal movement [61].

The above-mentioned polarization lability of macrophages is mediated by the ability to respond to changing levels of estrogen and estrogen-related factors. It ensures participation of macrophages in maintaining the homeostasis of female reproductive system during the ovarian-uterine cycle and pregnancy. The estrogen group includes several hormones: estradiol, estriol, and estrone [62,63]. A gradual increase in the estrogen blood levels during healthy pregnancy is mainly defined by increasing concentrations of estradiol [62]. At the beginning of pregnancy, estrogens are synthesized by the corpus luteum. From about the 9th week of gestation, the placenta becomes the main source of estradiol; it is produced predominantly by syncytiotrophoblast [63] and to a much lesser extent by Hofbauer cells [64,65]. The synthesis of estrogen in placenta depends on the adrenal glands of the mother and the fetus, since the placenta itself lacks some of the key enzymes of steroidogenesis [62]. Estrogens act on cells through two intracellular estrogen receptors ESR1 (ERα) and ESR2 (ERβ), as well as through G protein-coupled estrogen receptor 1 (GPER1). In human placental macrophages, expression of *Esr1* and *Gper1* is detectable, but ESR2 and the progesterone receptor are absent [66].

Although the estrogen receptors have been found in macrophages, it is believed that estrogens do not directly cause macrophage chemotaxis to the endometrium. It is rather that estrogen stimulates other cells (mostly fibroblasts) to produce cytokines which attract macrophages [44]. However, estrogens have multiple effects on macrophages. It is assumed that estradiol can stimulate macrophage proliferation directly or through other cells that produce mitogens EGF and IGF-1) [44]. Even in the absence of inflammatory mediators, estrogens cause the expression of early and late response genes in macrophages. In inflammation, estrogens promote polarization of macrophages to M2 phenotype and stimulate the synthesis of molecules involved in the extracellular matrix remodeling (proteases and their inhibitors) [67]. Estrogens can enhance or suppress the phagocytic ability of macrophages depending on the prevailing activation factors. Macrophages are capable of absorbing iron through the transferrin receptor 1 (TFRC) and CD163; estrogens enhance the absorption of iron ions through activation of TFRC, as well as by suppressing synthesis of hepcidin in the liver [44].

Some authors suggest that estrogens play a key role in the pathogenesis of PE, since they regulate angiogenesis and cause vasodilation [62]. A decrease in the level of estrogen in the blood of preeclamptic women is evident [68,69,70,71,72]. In PE, estradiol levels are decreased in the blood [69,71,73] and in the placenta [74]. Plasma concentrations of estrone and estriol in severe PE are also reduced, although in some of the studies such changes have not been detected [69,71]; there is also an evidence of reduced estriol levels in the placenta [70]. Androgens are another important group of hormones whose level is important for the normal course of pregnancy. It is supposed that they are responsible for cervical remodeling at term via regulation of cervical collagen fibril organization [75]. As for PE, it has been shown that testosterone level in women with PE is greatly increased [76]. It does not, however, answer the question of how it affects macrophages of placenta. Testosterone receptors have been found on the surface of macrophages and it was shown that the signal cascade triggered by testosterone includes the fluctuation of cytosolic calcium [77,78]. In another work, it was shown that androgens induce polarization of the lung macrophages towards the M2 phenotype [79]. 

Despite the large body of data concerning this topic, it is difficult to find a study addressing the role of estrogen- and androgen-dependent polarization of macrophages in PE. This area needs further research.

### 2.4. Monocytes in Preeclampsia

In considering the pathogenesis of PE, great emphasis is being made on oxidative stress and endothelial dysfunction occurring in the maternal body. Insufficient placentation causes abnormally regulated blood pressure in the maternal cardiovascular system, followed by blood supply shortages and, as a consequence, ischemia/reperfusion of the placenta. It is believed that under these conditions the hypoxic placenta synthesizes and secretes increased quantities of vasoactive substances promoting the release of a the number of signal factors such as placental debris, exosomes, microvesicles, cell-free nucleic acids, and pro-inflammatory cytokines into maternal blood flow. The presence of these markers is well described and highly symptomatic [80,81,82]. The situation eventually leads to a pronounced inflammatory response, oxidative stress and enhances apoptosis of placental cells [83]. Elevated levels of pro-inflammatory cytokines produced by various cell types lead to dramatic changes in the patterns of surface molecules of the endothelium and result in systemic endothelial dysfunction and subsequent hypertension [84,85,86]. Detailed reviews concerning the delicate immune balance in normal pregnancy and in PE are available from scientific databases [3,87]. 

Inflammation is a pronounced feature of PE; it involves cells of both adaptive and innate immunity. Due to the fact that monocytes circulate in the blood only for a few days, their quantity and composition reflect the severity of the patient’s clinical condition. Since generalized inflammation is a well-known feature of PE, changes in monocyte quantity and subset profile should be expected. Indeed, Wang and colleagues analyzed clinical records of more than three hundred patients with PE and found that in PE group the absolute monocyte count and the monocyte-lymphocyte ratio were significantly higher as compared with the control group (Table 1) [88]. As revealed by ROC-analysis, the monocyte-lymphocyte ratio has good diagnostic accuracy to distinguish between the normal condition and PE. In the work of Brien and colleagues, an increase in monocyte counts was also found typical for PE; the authors used CD14 marker to identify monocytes [89].

Characterization of monocyte subpopulations in PE became a subject of interest after 2010 [22,23]. Recent investigations confirm the previous findings on its relevance. In recent work of Alahakoon and colleagues, the authors estimated quantities of classical, intermediate and non-classical monocytes in blood samples from preeclamptic patients (with or without intrauterine growth restriction, IUGR) and uncomplicated pregnancies [92]. The authors observed a significantly lower content of classical monocytes for both PE groups (with or without IUGR), while the number of inflammatory monocytes which combined intermediate and non-classical subsets for these groups was significantly increased as compared with the control. Similar results were obtained by Jabalie et al. who observed a decline in the percentage of classical monocytes paralleled by an increase in the percentage of intermediate and non-classical monocytes in blood samples from preeclamptic women with or without metabolic syndrome [90].

Ma and colleagues analyzed cytokine profiles of serum from women with PE and also estimated the percentage of blood monocytes positive for M1 and M2 macrophage markers [93]. The counts of CD14+CD11c+CD163-(M1) monocytes in PE group were significantly increased, which correlated with the increased level of pro-inflammatory factors (IL-1, IL-6, and MCP-1). However, the works concerning M1 and M2 macrophage markers in the blood of women with PE are few, which indicates the necessity of further studies in this field. 

Several studies focused on the composition of umbilical cord blood in PE have obvious scientific novelty since fetal participation is rarely considered in the context of PE. Interestingly, the authors come to the same observation: a significant reduction in the classical monocyte subset and a significant increase in non-classical monocyte subset were observed for the cord blood in PE group [91].

Summarizing these data leads to a general conclusion that the observed PE-associated changes in counts and composition of blood monocytes towards the prevalence of non-classical subset indicate progression of the inflammation symptoms in the maternal organism. The upheaval of inflammatory reaction during PE is possibly caused by extracellular agents, which appear in blood, and cytokines, which activate monocytes [103]. Since monocyte counts are included in routine clinical blood tests, and given that phenotyping of monocytes by flow cytometry is a straightforward procedure, appearance of monocyte-based tests for PE prediction as a routine practice may be expected. Surely it would require a prospective study to assess the prognostic value of monocyte profile indicators in the blood of a pregnant woman who would have PE and absolute standardization of all manipulations. To date, none of the existing tests reliably evaluates the risks of PE. At present, only a few markers associated with PE, such as endoglin, placental growth factor (PlGF) and sFlt-1 (soluble fms-like tyrosine kinase 1), have been sufficiently studied.

### 2.5. Macrophages in Preeclampsia

The increase in non-classical monocyte subset may affect the composition of tissue macrophages in the endometrium and be responsible for the poor placentation in PE [104]. Appropriate balance between pro- and anti-inflammatory macrophages in the placenta is essential for healthy pregnancy and its optimal outcome. It has been suggested that transition to the M2 profile, which normally occurs in the second trimester, is blocked in PE; more specifically, it is canceled at the early stages of the disease [105]. As a consequence, M1 responses remain unsuppressed, and cytokines exhibit a pro-inflammatory profile with elevated levels of IFN-γ, TNFalpha, IL6 and reduced levels of IL-4 and IL-10 [105,106].

Behavior of resident placental macrophages in PE has not received the proper attention of the researchers as yet. This can be possibly explained by the complexity of the biomaterial collection and the time-consuming procedure of isolation and phenotyping of the cells from this material in contrast to the easily obtained blood samples. However, this subject requires a comprehensive study. In the context of any pregnancy complication, placental macrophages should be considered as two populations: Hofbauer cells of the fetal placenta and decidual macrophages of the maternal placental part.

Recently published works comprise somewhat controversial numerical estimates for both Hofbauer cells and decidual macrophages in preeclamptic placentas. Yang and colleagues observed significantly lower numbers of CD14+ Hofbauer cells in PE placenta as compared with the healthy control but no corresponding significant difference in CD68+ cell numbers was found [94]. Tang et al. observed significantly declined numbers of CD68+ Hofbauer cells in PE group in comparison with the gestation age-matched preterm birth control group [95]. By contrast, Evsen and colleagues, on the contrary, observed increased Hofbauer cell numbers in PE complicated by HELLP syndrome group compared to the control group; the authors also used CD68 as a macrophage marker [96]. Saeed et al. report a two-fold increase in the Hofbauer cell number in preeclamptic placentas in comparison to normotensive pregnancy [98]. As for decidual macrophages, their comparative abundance in the preeclamptic placenta is also a controversial subject. Schonkeren and colleagues reported an increase in number of CD14+ cells in the decidua basalis for preterm preeclamptic pregnancies compared with preterm control pregnancies [50]. Milosevic-Stevanovic et al. observed higher numbers of CD68+ decidual cells in PE compared to healthy control placentas [100]. In one study, a significant increase in number of CD68+ cells, in both fetal and decidual parts of placenta, was observed in preeclamptic group as compared to controls [97]. At the same time, several research groups report that decidual macrophage numbers in PE placentas are reduced [94,101,102]. Apparent inconsistencies between the studies may be explained by the use of different technical approaches (cell markers, antibodies, signal detection protocols, etc.) and the difference in the formation of studied groups.

The issue of macrophage polarization in preeclamptic placenta looks less ambiguous. Yang and colleagues showed that the level of CD163+ Hofbauer cells is significantly downregulated in PE compared with healthy pregnancies [94]. In work of Tang’s and colleagues they also observed a decrease in CD163+ cell numbers compared with the preterm control [95]. Przybyl et al. reported reduced CD74+ cell numbers in preeclamptic placentas; according to the proposed model, this may lead to a pro-inflammatory signature [99]. Ma et al. observed an increase in the percentage of CD11b-iNOS co-labeled cells and a concomitant decrease in the percentage of CD11b-Arg1 co-labeled cells in preeclamptic placentas as compared with normal ones [93]; the combinations of markers reflect M1 and M2 phenotypes, respectively. The ratio of CD163/CD14 decidual cells was also found to be declined in placental samples collected from women with PE [50].

The shift in the balance of M2/M1 macrophages towards M1 is explained by high levels of pro-inflammatory cytokines and low levels of anti-inflammatory cytokines within the preeclamptic placenta [107,108]. In addition to the altered cytokine production, there is also a cellular axis in the process of polarization. A number of studies suggest an essential role of placental mesenchymal stem cells in macrophage polarization and their ability to affect their activation [109,110]. Wang and colleagues revealed the role of hyaluronan in maintaining normal pregnancy. Their findings indicate that high levels of hyaluronan induce M2 polarization and regulate production of cytokines (e.g., IL-10) by decidual macrophages [111].

The summarizing scheme is presented in Figure 1. Despite the large body of available data, several questions are still remaining unanswered. When do the observed changes in macrophage polarization really emerge—at the early stages of pregnancy or after the PE manifestation? At what gestational age could they be valid as markers? Can we use macrophages and monocytes for therapeutic purposes?

### 2.6. Potential Therapeutic Approaches

Reprogramming of macrophages seems to be an attractive therapeutic strategy. A number of FDA approved approaches involving cellular and gene therapy are now used in clinical practice [112]. Macrophages derived from monocytes can be activated in different ways by varying combinations of external stimuli. The ex vivo reprogramming of macrophages conventionally aims to polarize them towards the anti-inflammatory phenotype in order to make the M2 polarized macrophages confront inflammation in maternal body. The idea of ex vivo reprogramming of autologous macrophages has been developed since the 1980s [113]. By now, the reprogrammed macrophages have been successfully used in a number of therapeutic cases including treatment of cancer, transplantation, and stimulation of regeneration [114,115,116]. Common approaches for the ex vivo macrophage polarization are stimulation of the cells with cytokine cocktails, genetic manipulation, or using specific low-molecular inhibitors of transcription factors [117,118]. iPS-ML, the macrophage-like myelomonocytic cells generated from the human induced pluripotent stem cells, are also amenable to ex vivo polarization [119]. Injection of the autologous monocyte-derived M2-polarized macrophages at a certain time of gestation (or at the stage of its planning by taking into account the risks of PE) may evolve into a new strategy for PE treatment. Such therapy seems to be promising due to reports about the absence of adverse reactions and long-term side effects after macrophages transplantation in other diseases [120,121]. A possible side effect of the proposed therapy may be the phenomenon of maternal–fetal cellular trafficking—the ability of mother and fetus cells pass the placental barrier [122]. The presence of fetal cells in the maternal circulation is known as fetal microchimerism, while the presence maternal cells in the fetal organism is known as maternal microchimerism. Indeed, in a number of works it was shown that fetuses with severe congenital diaphragmatic hernia have increased levels of maternal microchimerism [122,123]. However, this does not mean that the activated *ex vivo* auto-monocytes will necessarily penetrate the placenta. This question has not been sufficiently investigated.

## 3. Summary and Conclusions

Compared to other pregnancy complications, PE is the main cause of maternal morbidity and mortality. For several decades, researchers have been trying to understand the causes of PE, considering the disorder from different points of view. An important aspect of PE development is the response of maternal innate immunity to various proinflammatory stimuli from the placenta. Normally, the maternal organism adapts to the presence of the fetus; however, at PE, a pronounced inflammatory process occurs. The consequences of this process are manifested in the increased monocyte numbers and altered subset composition, including changes in counts and polarization of placental macrophages. The recent examples of observations listed in this review suggest that monitoring of blood composition and phenotyping of monocytes over the course of pregnancy might be considered as screening tests for PE. However, such tests must be used in combination with other PE prognostic markers because the symptoms of some other complications and conditions (e.g., stress, infections or cancer) occasionally could mimic the PE-associated monocyte changes [8]. Monocytes and tissue macrophages are the extremely important cell types involved in PE pathogenesis. Their possible application as PE predictors and/or therapeutics agents holds great promise.

## Figures and Tables

**Figure 1 ijms-20-03695-f001:**
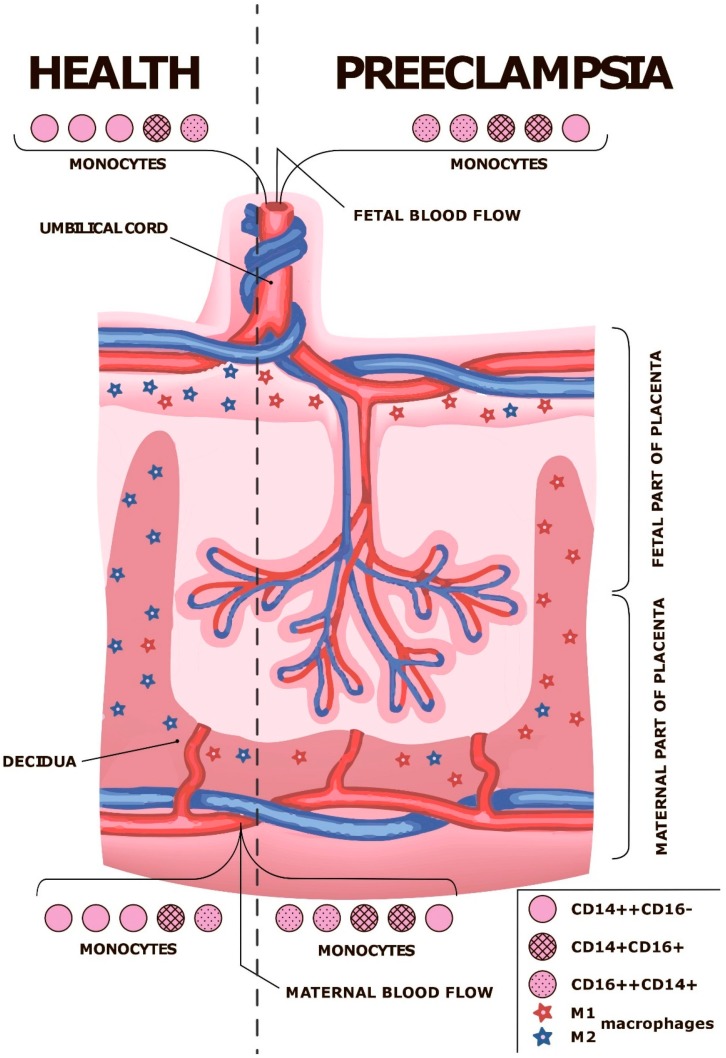
Summarizing scheme. Monocyte–macrophage system in decidua, fetal and maternal placental parts and blood flow of mother and fetus. Classical (CD14++CD16−), non-classical (CD16++CD14+) and intermediate (CD14+CD16+) monocyte populations as well as proinflammatory (M1) and anti-inflammatory (M2) macrophages are shown. Modified from [26] (distributed under CC-BY).

**Table 1 ijms-20-03695-t001:** Summary observations concerning monocyte–macrophage system in preeclampsia (PE).

Subject	Observation in PE	Quantity: Control vs. PE	Reference
Monocyte	↑ Monocyte count ↑ Monocyte-lymphocyte ratio	161/302	[88]
	↑ Monocyte count	20/20	[89]
	↓ CD14++CD16−, ↑ CD14+CD16++	11/17	[23]
		40/35	[90]
		8/4 (umbilical cord blood)	[91]
	↓ CD14++CD16−; ↑ (CD14+CD16++ and CD14++CD16+)	24/9	[92]
		23/26	[22]
	↑ CD14+CD11c+CD163-	30/22	[93]
Macrophages in placenta			
Hofbauer cells	↓ CD14+	30/10	[94]
	↓ CD68+	11/10	[95]
	↑ CD68+	20/20	[96]
		6/6	[97]
	↑ Hofbauer cells number	50/50	[98]
	↓ CD163+	30/10	[94]
	↓ CD163+	11/10	[95]
	↓ CD74+	28/24	[99]
	↓ CD11b+Arg1+ ↑ CD11b+ iNOS+	22/30	[93]
Decidual macrophages	↑ CD14+	5/6	[50]
	↑ CD68+	20/30	[100]
		6/6	[97]
	↓ CD14+	12/12	[101]
		6/6	[102]
	↓ CD163/CD14	5/6	[50]

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
