# Peer review of "Role of the Monocyte–Macrophage System in Normal Pregnancy and Preeclampsia"

_ijms, 2019, doi:10.3390/ijms20153695_

Round 1

Reviewer 1 Report

The review is well written and presents comprehensive update about the role of the monocyte-macrophage system in normal and preeclamptic pregnancy, however, following points need attention-

1- In preeclamptic pregnancy not only estradiol but androgen levels are also altered. It has been shown through various reports that testosterone levels are increased in PE and sometimes linked with the severity of the PE. It will be valuable to mention/ discuss the effect of increased androgens on monocyte-macrophage system during normal and preeclamptic pregnancies.

2- Figures (on first page and figure 1) needs detailed figure legends for better comprehension.

2- in figure- 1 "HEALTH" is mentioned on the top left, which needs to be corrected.

Author Response

Thank you for considering our work. All your comments have been addressed, with corresponding changes made directly to the manuscript where appropriate. It is our belief that the manuscript is substantially improved after making the suggested edits.

1- In preeclamptic pregnancy not only estradiol but androgen levels are also altered. It has been shown through various reports that testosterone levels are increased in PE and sometimes linked with the severity of the PE. It will be valuable to mention/ discuss the effect of increased androgens on monocyte-macrophage system during normal and preeclamptic pregnancies.

Thank you for this important comment. We added the fragment concerning androgen at preeclampsia and androgen-depending polarization in the text (Line 250-257, Page 6).

2- Figures (on the first page and figure 1) needs detailed figure legends for better comprehension.

Thank you for this point. The first picture (graphical abstract) should be «self-explanatory image» as written in requirements. So, we modified it in accordance to your comment to make it clearer. As for Fig.1 in the body of the Manuscript, in the revised version it is replaced by more comprehensive image and additional description was added (Page 10).

3- in figure- 1 "HEALTH" is mentioned on the top left, which needs to be corrected.

Thank you. We corrected it in the new version of the picture.

Reviewer 2 Report

The review article describes the monocyte macrophage system in reproductive organ, pregnancy, and preeclampsia. The objective of this review is to summarize what is known about the role of monocytes and macrophages in preeclampsia as well as describe knowledge gaps in this area. Finally, potential therapeutic strategies centered around these altered cell populations are discussed.

Major Comments:

Lines 121-122- The M1/M2 balance can be changed as described by the authors. However, one might consider that suppression of the M2 pathway or stimulation of the M1 pathway can also change the balance. As written, the sentence suggests that M1 polarization is the default phenotype of macrophages unless the M1 activation pathway is suppressed or the M2 activation pathways are stimulated. Is this what the authors are suggesting?

Lines 123-125- It is unclear what M1/M2 balance shift the authors are attempting to achieve. Is this an attempt to increase or reduce the M1/M2 ratio? Does blockade of IL-6 cause activation of the M1 macrophage polarization or suppression? This sentence should be revised.

Lines 173-184: A figure to accompany the description of macrophage roles in section 2.3 is needed and would greatly enhance the readability of the article.

Figure 1: The figure suggests that M2 cells are increased in PE compared to a healthy pregnancy. This is contradictory to the text in lines 355-357

Section 2.6: The authors should discuss any potential risks with the proposed therapeutic strategies. Have any of the uses described for other diseases been examined in this vulnerable population? What are the potential effects of these therapies on the fetus? Would the cells potentially cross the placenta?

Minor/Specific Comments:

Line 41: change “various cell types of blood and tissues” to “various cell types in blood and tissues”

Line 112: add “mice” at the end of the sentence

Line 219: estrone is repeated in the list of estrogen hormones

Line 254: remove “inadequate”

Lines 257-259: Sentence is unclear. Perhaps change to “vasoactive substances promoting the release of a number of signaling factors such as placental debris……and pro-inflammatory cytokines into the maternal circulation

Line 344: What do the authors mean by “groups formation”?

Lines361-362: Specifiy is hyaluronan promotes or inhibits M2 polarization and what type of cytokines (pro or anti-inflammatory) are secreted by decidual cells after exposure to hyaluronan.

Author Response

Thank you for considering our work. All your comments have been addressed, with corresponding changes made directly to the manuscript where appropriate. It is our belief that the manuscript is substantially improved after making the suggested edits.

Major Comments:

Lines 121-122- The M1/M2 balance can be changed as described by the authors. However, one might consider that suppression of the M2 pathway or stimulation of the M1 pathway can also change the balance. As written, the sentence suggests that M1 polarization is the default phenotype of macrophages unless the M1 activation pathway is suppressed or the M2 activation pathways are stimulated. Is this what the authors are suggesting?

Yes, you are right, the sentence is not clear. We replaced it with the following: "Macrophages could be polarized using specific inductors (what is often called the direct polarization), or using an indirect method: blockage the undesired phenotype and getting the reverse, desired, phenotype". (Lines 121-123, Page 3).

Lines 123-125- It is unclear what M1/M2 balance shift the authors are attempting to achieve. Is this an attempt to increase or reduce the M1/M2 ratio? Does blockade of IL-6 cause activation of the M1 macrophage polarization or suppression? This sentence should be revised.

Thank you for this point. According to your recommendation, we revised the sentence to: «Thus, the in vivo M2 phenotype could be achieved by blocking IL-6-signaling [15] or by adding IL-4 [18] and IL-10 what induced М2а and M2c polarization, respectively [13]». (Lines 125-127, Page 3).

Lines 173-184: A figure to accompany the description of macrophage roles in section 2.3 is needed and would greatly enhance the readability of the article.

We replaced Fig.1 by it`s new version, which summarized the data from the text. We hope that this illustration will be useful for understanding section 2.3. (Page 10).

Figure 1: The figure suggests that M2 cells are increased in PE compared to a healthy pregnancy. This is contradictory to the text in lines 355-357

Thank you. Yes, you are right, we corrected this mistake in the new version of the picture. Figure 1 was replaced by a more comprehensive image and additional description was added (Page 10).

Section 2.6: The authors should discuss any potential risks with the proposed therapeutic strategies. Have any of the uses described for other diseases been examined in this vulnerable population? What are the potential effects of these therapies on the fetus? Would the cells potentially cross the placenta?

Thank you for your comment. Your question raises the important topic. We added the fragment concerning possible side effects and the phenomenon of the maternal-fetal cellular trafficking in text. (Lines 399-408, Page 11).

Minor/Specific Comments:

Line 41: change “various cell types of blood and tissues” to “various cell types in blood and tissues”

We did it, thank you.

Line 112: add “mice” at the end of the sentence

Fixed.

Line 219: estrone is repeated in the list of estrogen hormones

The mistake is corrected.

Line 254: remove “inadequate”

Fixed, thank you.

Lines 257-259: Sentence is unclear. Perhaps change to “vasoactive substances promoting the release of a number of signaling factors such as placental debris……and pro-inflammatory cytokines into the maternal circulation

We did it as you recommended, thank you.

Line 344: What do the authors mean by “groups formation”?

We mean the different clinical and demographic data used for the formation of studied groups. We corrected this sentence. (Line 354, Page 8).

Lines361-362: Specifiy is hyaluronan promotes or inhibits M2 polarization and what type of cytokines (pro or anti-inflammatory) are secreted by decidual cells after exposure to hyaluronan.

In Wang`s work, they confirmed that hyaluronan promotes M2 polarization and IL-10 production in decidual macrophages. We added it in the text as you recommended. (Lines 371-372, Page 8).